# The ReprGesture entry to the GENEA Challenge 2022

SICHENG YANG, Tsinghua University, China

ZHIYONG WU*, Tsinghua University, China and The Chinese University of Hong Kong, China

MINGLEI LI*, Huawei Cloud Computing Technologies Co., Ltd, China

MENGCHEN ZHAO, Huawei Noah's Ark Lab, China

JIUXIN LIN, LIYANG CHEN, and WEIHONG BAO, Tsinghua University, China

This paper describes the ReprGesture entry to the Generation and Evaluation of Non-verbal Behaviour for Embodied Agents (GENEA) challenge 2022. The GENEA challenge provides the processed datasets and performs crowdsourced evaluations to compare the performance of different gesture generation systems. In this paper, we explore an automatic gesture generation system based on multimodal representation learning. We use WavLM features for audio, FastText features for text and position and rotation matrix features for gesture. Each modality is projected to two distinct subspaces: modality-invariant and modality-specific. To learn inter-modality-invariant commonalities and capture the characters of modality-specific representations, gradient reversal layer based adversarial classifier and modality reconstruction decoders are used during training. The gesture decoder generates proper gestures using all representations and features related to the rhythm in the audio. Our code, pre-trained models and demo are available at https://github.com/YoungSeng/ReprGesture.

CCS Concepts: • **Computing methodologies** → **Artificial intelligence**; **Natural language processing**; • **Human-centered computing** → **Human computer interaction (HCI)**.

Additional Key Words and Phrases: gesture generation, data-driven animation, modality-invaiant, modality-specific, representation learning, deep learning

**ACM Reference Format:**

Sicheng Yang, Zhiyong Wu, Minglei Li, Mengchen Zhao, Jiuxin Lin, Liyang Chen, and Weihong Bao. 2022. The ReprGesture entry to the GENEA Challenge 2022. In *INTERNATIONAL CONFERENCE ON MULTIMODAL INTERACTION (ICMI '22), November 7–11, 2022, Bengaluru, India.* ACM, New York, NY, USA, 8 pages. https://doi.org/10.1145/3536221.3558066

## 1 INTRODUCTION

Nonverbal behavior plays a key role in conveying messages in human communication [13], including facial expressions, hand gestures and body gestures. Co-speech gestures introduce better self-expression. In the virtual world, it helps to present a rather realistic digital avatar. Gesture generation studies how to generate human-like, natural, speech-oriented gestures. There are many different techniques for gesture generation. In this paper, we focus on the task of speech-driven gesture generation. Representative speech-driven gesture generation are either rule-based or data-driven [19].

Many data-driven works for gesture generation are based on multimodal fusion and representation learning. Taras et al. map speech acoustic and semantic features into continuous 3D gestures [12]. Youngwoo et al. propose an end-to-end model to generate co-speech gestures using text, audio, and speaker identity [19]. Jing et al. sample gesture

---

*Corresponding authors

Manuscript submitted to ACM

in a variational autoencoder (VAE) latent space and infer rhythmic motion from speech prosody to address the non-deterministic mapping from speech to gesture [18]. Taras et al. propose a speech-driven gesture-production method based on representation learning [11]. Xian et al. propose the hierarchical audio features extractor and pose inferrer to learn discriminative representations [17]. Jing et al. present a co-speech gesture generation model whose latent space is split into shared code and motion-specific code [15].

However, gesture generation is a challenging task because of cross-modality learning issue and the weak correlation between speech and gestures. The inherent heterogeneity of the representations creates a gap among different modalities. It is necessary to address the weak correlation among different modalities and provide a holistic view of the multimodal data during gesture generation.

Inspired by [19] and [6], we propose a gesture generation system based on multimodal representation learning. In particular, we first extract features of audio, text and gestures. Then, a system consisting of four components is proposed: (1) Each modality is projected to two distinct representations: modality-invariant and modality-specific. (2) A gradient reversal layer based adversarial classifier is used to reduce the discrepancy between the modality-invariant representations of each modality. (3) Modality decoders are used to reconstruct each modality, allowing modality-specific representations to capture the details of their respective modality. (4) The gesture decoder takes six modality representations (two per modality) and rhythm-related features in audio as its input and generates proper gestures.

The main contributions of our work are: (1) A multimodal representation learning approach is proposed for gesture generation, which ensures comprehensive decoupling of multimodal data. (2) To solve the problem of heterogeneity of different modalities in feature fusion, each modality is projected to two subspaces (modality-invariant and modality-specific) to get multimodal representations using domain learning and modality reconstruction. (3) Ablation studies demonstrate the role of different components in the system.

The task of the GENEA 2022 challenge is to generate corresponding gestures from the given audio and text. A complete task description can be accessed in [21]. We submitted our system to the GENEA 2022 challenge to be evaluated with other gesture generation systems in a large user study.

## 2 METHOD

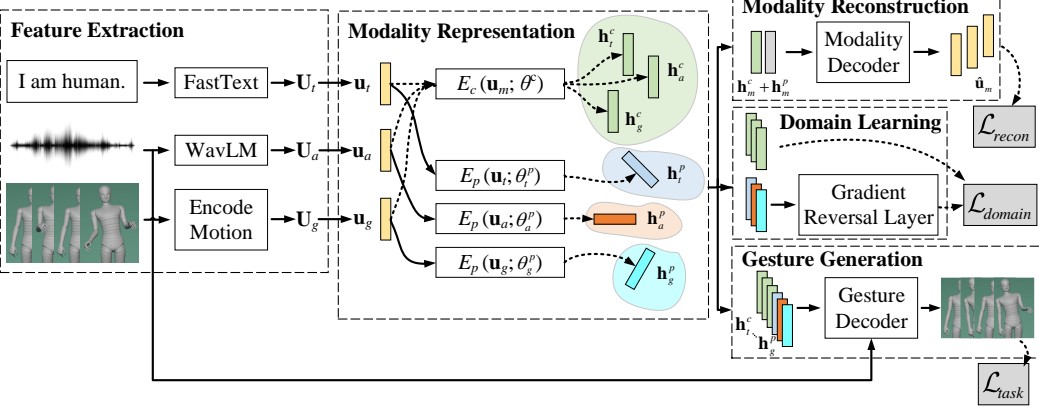

Fig. 1. Gesture generation through modality -invariant and -specific subspaces.

### 2.1 The architecture of the proposed system

As shown in Figure 1, the system generates a sequence of human gestures from a sequence of $\mathbf{u}_m (m \in \{t, a, g\})$ that contain the features of text, audio and seed gestures. The architecture of the proposed model consists of five modules: feature extraction, modality representation, modality reconstruction, domain learning and gesture generation. The following describes each of these modules in detail.

#### 2.1.1 Feature extraction.

For each of the modality, the pipeline of extracting features is as follows:

- Text: We first use FastText [1] to get the word embeddings. Padding tokens are inserted to make the words temporally match the gestures by following [19]. One-dimensional (1D) convolutional layers are then adopted to generate 32-D text feature sequence $\mathbf{U}_t$ ('$t$' for 'text') from the 300-D word embeddings.
- Audio: All audio recordings are downsampled to 16kHz, and features are generated from the pre-trained models of WavLM Large [3]. We further adjust sizes, strides and padding in the 1D convolutional layers to reduce the dimension of features from 1024 to 128 forming the final audio feature sequence $\mathbf{U}_a$ ('$a$' for 'audio').
- Gesture: Due to the poor quality of hand motion-capture, we only use 18 joints corresponding to the upper body without hands or fingers. Root normalization is used to make objects face the same direction. We apply standard normalization (zero mean and unit variant) to all joints. Seed gestures for the first few frames are utilized for better continuity between consecutive syntheses, as in [19]. On top of these, position and 3 × 3 rotation matrix features are computed, and the size of final gesture sequence $\mathbf{U}_g$ ('$g$' for 'gesture') feature is 216.

#### 2.1.2 Modality representation.

First, for each modality $m \in \{t, a, g\}$, we use a linear layer with leaky ReLU activation and layer normalization to map its feature sequence $\mathbf{U}_m$ into a new feature sequence $\mathbf{u}_m \in \mathbb{R}^{T \times d_h}$ with the same feature dimension $d_h$. Then, we project each sequence $\mathbf{u}_m$ to two distinct representations: modality-invariant $\mathbf{h}_m^c$ and modality-specific $\mathbf{h}_m^p$. Afterwards, $\mathbf{h}_m^c$ learns a shared representation in a common subspace with distributional similarity constraints [5]. $\mathbf{h}_m^p$ captures the unique characteristics of that modality. We derive the representations using the simple feed-forward neural encoding functions:

$$\mathbf{h}_m^c = E_c \left( \mathbf{u}_m; \theta^c \right), \quad \mathbf{h}_m^p = E_p \left( \mathbf{u}_m; \theta_m^p \right) \tag{1}$$

Encoder $E_c$ shares the parameters $\theta^c$ across all three modalities, whereas $E_p$ assigns separate parameters $\theta_m^p$ for each modality.

#### 2.1.3 Representation learning.

Domain learning can improve a model's ability to extract domain-invariant features [2]. We use an adversarial classifier to minimize domain loss that reduces the discrepancy among shared representations of each modality.

The domain loss can be formulated as:

$$\mathcal{L}_{domain} = -\sum_{i=1}^{3} \mathbb{E}[\log \left( D_{repr}(d_m) \right)] \tag{2}$$

where $D_{repr}$ represents feed-forward neural discriminator, $d_m$ represents the result after gradient reversal of $\mathbf{h}_m^p$.

The modality reconstruction loss $\mathcal{L}_{recon}$ is computed on the reconstructed modality and the original input $\mathbf{u}_m$. The $\mathcal{L}_{recon}$ is used to ensure the hidden representations to capture the details of their respective modality.

Specifically, a modality decoder $D$ is proposed to reconstruct $\mathbf{u}_m$:

$$\hat{\mathbf{u}}_m = D \left( \mathbf{h}_m^c + \mathbf{h}_m^p; \theta^d \right) \tag{3}$$

where $\theta^d$ are the parameters of the modality decoder. The modality reconstruction loss can then be computed as:

$$\mathcal{L}_{\text{recon}} = \frac{1}{3}\left(\sum_{m\in\{t,a,g\}} \frac{\|\mathbf{u}_m - \hat{\mathbf{u}}_m\|_2^2}{d_h}\right) \tag{4}$$

where $\|\cdot\|_2^2$ is the squared $L_2$-norm.

### 2.1.4 Gesture generation.

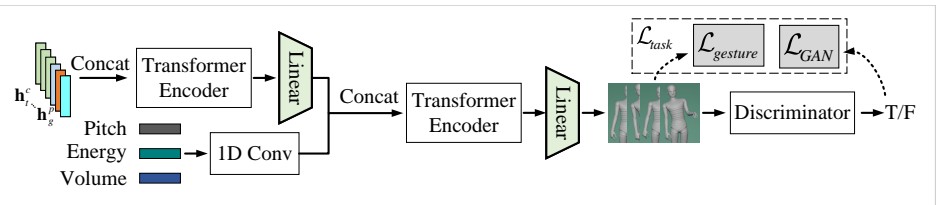

Fig. 2. Architecture of the gesture generation module.

We use generative adversarial network (GAN) based gesture decoder for generating gestures. Gestures are directly related to rhythm and beat, thus we concatenate audio rhythm related features (pitch, energy and volume) and the output of six stacked modality representations together and send them to Transformer encoders with multi-head self-attention as the generator, as shown in Figure 2. The generator part is trained using $\mathcal{L}_{gesture}$ consisting of the Huber loss and the MSE loss, and the discriminator part is trained with $\mathcal{L}_{GAN}$.

$$\mathcal{L}_{gesture} = \alpha \cdot \mathbb{E}\left[\frac{1}{t}\sum_{i=1}^{t}\text{HuberLoss}\,(g_i,\hat{g}_i)\right] + \beta \cdot \mathbb{E}\left[\frac{1}{t}\sum_{i=1}^{t}\|\,(g_i,\hat{g}_i)\,\|_2^2\right] \tag{5}$$

$$\mathcal{L}_{GAN} = -\mathbb{E}[\log(D_{gesture}(g))] - \mathbb{E}[\log(1 - D_{gesture}(\hat{g}))] \tag{6}$$

where $D_{gesture}$ represents gesture discriminator using multilayered bidirectional gated recurrent unit (GRU) [4] that outputs binary output for each time step, $t$ is the length of the gesture sequence, $g_i$ represents the $i$th human gesture, $\hat{g}_i$ represents the $i$th generated gesture.

The loss of the proposed system can be computed as:

$$\mathcal{L}_{total} = \mathcal{L}_{gesture} + \gamma \cdot \mathcal{L}_{GAN} + \delta \cdot \mathcal{L}_{domain} + \epsilon \cdot \mathcal{L}_{recon} \tag{7}$$

## 2.2 Data processing and experiment setup

### 2.2.1 Data and data processing.

In the challenge, the Talking With Hands 16.2M [14] is used as the standard dataset. Each video is separated into two independent sides with one speaker each. The audio and text in the dataset have been aligned. For more details please refer to the challenge paper [21]. We note that the data in the training, validation and test sets are extremely unbalanced, so we only use the data from the speaker with identity "1" for training. And we believe that if speech and gesture data are trained on the same person, the gesture behavior would match the speech.

### 2.2.2 Experiment setup.

The proposed system is trained on training data only, using the ADAM [9] optimizer (learning rate is e-4, $\beta_1 = 0.5$, $\beta_2 = 0.98$) with a batch size of 128 for 100 steps. We set $\alpha = 300$, $\beta = 50$ for Equation (5) and $\gamma = 5, \delta = 0.1, \epsilon = 0.1$ (we

noticed in our experiments that too large $\delta$ and $\epsilon$ will lead to non-convergence) for Equation (7). There is a warm-up period of 10 epochs in which the $\mathcal{L}_{GAN}$ is not used ($\gamma = 0$). The feature dimension $d_h$ of sequence $\mathbf{u}_m$ is 48. During training, each training sample having 100 frames is sampled with a stride of 10 from the valid motion sections; the initial 10 frames are used as seed gesture poses and the model is trained to generate the remaining 90 poses (3 seconds).

## 3 EVALUATION

### 3.1 Evaluation setup

The GENEA Challenge 2022 evaluation is divided into two tiers, and we participated in the upper-body motion tier. The challenge organizers conducted a detailed evaluation comparing all submitted systems[21]. The challenge evaluates human-likeness to assess motion quality, and appropriateness to assess how well the gestures match the speech. The evaluation is based on the HEMVIP methodology [8] and Mean Opinion Score (MOS) [7]. There are in total 11 systems participated in the upper-body tier. The following abbreviations are used to represent each model in the evaluation:

- UNA: Ground truth ('U' for the upper-body tier, 'NA' for 'natural').
- UBT: The official text-based baseline [20], which takes transcribed speech text with word-level timing information as the input modality ('B' for 'baseline', 'T' for 'text').
- UBA: The official audio-based baseline [10], which takes speech audio into account when generating output ('A' for 'audio').
- USJ–USQ: 8 participants' submissions to the upper-body tier (ours is USN).

For more details about the evaluation studies, please refer to the challenge paper [21].

### 3.2 Subjective evaluation results and discussion

#### 3.2.1 Human-likeness Evaluation.

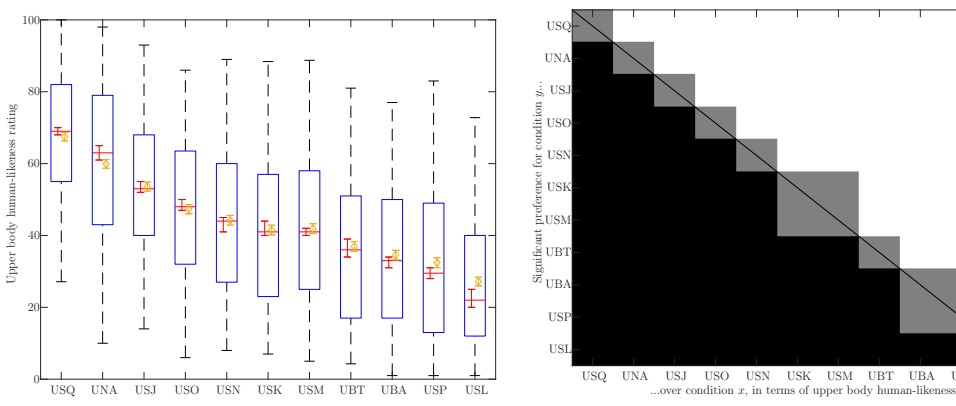

(a) Box visualizing the ratings distribution in Upper-body study.

(b) Significance of pairwise differences between conditions.

Fig. 3. (a) Red bars are the median ratings (each with a 0.05 confidence interval); yellow diamonds are mean ratings (also with a 0.05 confidence interval). Box edges are at 25 and 75 percentiles, while whiskers cover 95% of all ratings for each condition. (b) White means that the condition listed on the $y$-axis rated significantly above the condition on the $x$-axis, black means the opposite ($y$ rated below $x$), and grey means no statistically significant difference at the level $\alpha = 0.05$ after Holm-Bonferroni correction.

In this evaluations, study participants are asked to rate "How human-like does the gesture motion appear?" on a scale from 0 (worst) to 100 (best). Bar plots and significance comparisons are shown in Figure 3. Our system (USN) receives a median score of 44 and a mean score of 44.2, and is ranked fourth among the participating systems.

### 3.2.2 Appropriateness evaluation.

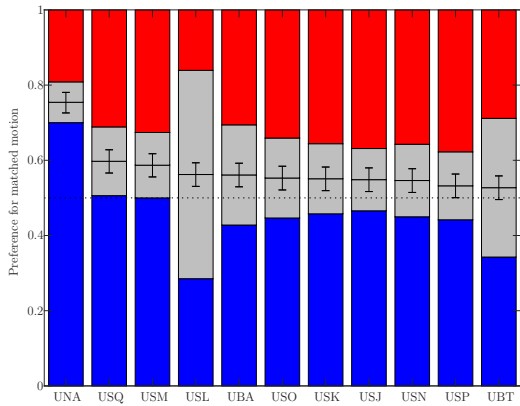

Fig. 4. Bar plots visualizing the response distribution in the appropriateness studies. The blue bar (bottom) represents responses where subjects preferred the matched motion, the light grey bar (middle) represents tied ("They are equal") responses, and the red bar (top) represents responses preferring mismatched motion, with the height of each bar being proportional to the fraction of responses in each category. The black horizontal line bisecting the light grey bar shows the proportion of matched responses after splitting ties, each with a 0.05 confidence interval. The dashed black line indicates chance-level performance.

In this evaluation, participants are asked to choose the character on the left, on the right, or indicate that the two are equally well matched to response "Please indicate which character's motion best matches the speech, both in terms of rhythm and intonation and in terms of meaning." Bar plots are shown in Figure 4. Our system (USN) receives a "Percent matched" 54.6, which identifies how often participants preferred matched over mismatched motion in terms of appropriateness. Our system is rated seventh in appropriateness among the participants' submissions. It should be noted that the difference of our system to the five higher-ranked systems (USL, UBA, USO, USK and USJ) is not significant. Furthermore, if we only consider the ratio of matched motion, i.e., the blue bar in Figure 4, our system is ranked fifth among the participating systems.

## 3.3 Ablation studies

Moreover, we conduct ablation studies to address the performance effects from different components in the system. The GENEA challenge computes some objective metrics of motion quality by GENEA numerical evaluations[1]. For calculation and meaning of these objective evaluation metrics, please refer to the challenge paper [21]. A perfect natural system should have average jerk and acceleration very similar to natural motion. The closer the Canonical correlation analysis (CCA) to 1, the better. Lower Hellinger distance and Fréchet gesture distance (FGD) are better. To compute the FGD, we train an autoencoder using the training set of the challenge.

The results of our ablations studes are summarized in Table 1. Supported by the results, when we do not use WavLM to extract audio features, but use 1D convolution instead, the Hellinger distance average and FGD on feature space present

---

[1]https://github.com/genea-workshop/genea_numerical_evaluations

Table 1. Ablation studies results. 'w/o' is short for 'without'. Bold indicates the best metric, i.e. the one closest to the ground truth.

| Name | Average jerk | Average acceleration | Global CCA | CCA for each sequence | Hellinger distance average ↓ | FGD on feature space ↓ | FGD on raw data space ↓ |
|---|---|---|---|---|---|---|---|
| Ground Truth (GT) | $18149.74 \pm 2252.61$ | $401.24 \pm 67.57$ | 1.000 | $1.00 \pm 0.00$ | 0.0 | 0.0 | 0.0 |
| ReprGesture | $2647.59 \pm 1200.05$ | $146.90 \pm 46.09$ | 0.726 | $\mathbf{0.95 \pm 0.02}$ | **0.155** | 0.86 | **184.753** |
| w/o WavLM | $1775.09 \pm 512.08$ | $77.53 \pm 21.92$ | **0.761** | $0.94 \pm 0.03$ | 0.353 | 3.054 | 321.383 |
| w/o $\mathcal{L}_{GAN}$ | $\mathbf{9731.54 \pm 3636.06}$ | $\mathbf{242.15 \pm 81.81}$ | 0.664 | $0.93 \pm 0.03$ | 0.342 | 2.053 | 277.539 |
| w/o $\mathcal{L}_{recon}$ | $533.95 \pm 193.18$ | $39.49 \pm 12.23$ | 0.710 | $0.93 \pm 0.03$ | 0.283 | 0.731 | 659.150 |
| w/o $\mathcal{L}_{domain}$ | $2794.79 \pm 1153.75$ | $135.62 \pm 25.13$ | 0.707 | $0.94 \pm 0.03$ | 0.267 | **0.653** | 874.209 |
| w/o Repr | $2534.34 \pm 1151.38$ | $123.02 \pm 40.90$ | 0.723 | $0.94 \pm 0.04$ | 0.298 | 0.829 | 514.706 |

the worst performance. When the model is trained without the GAN loss, the average jerk and average acceleration are better, but the global CCA and CCA for each sequence are decreased. When the reconstruction loss is removed, the average jerk and average acceleration are worst. The generated gesture movements are few and of small range. When the model is trained using Central Moment Discrepancy (CMD) loss [6] instead of domain loss, the best FGD on feature space and the worst FGD on raw data space are obtained. When the modality representations are removed (w/o Repr), we feed the modality sequence $\mathbf{u}_t, \mathbf{u}_a$ and $\mathbf{u}_g$ directly to the gesture decoder and only use the $\mathcal{L}_{task}$ loss, the performances of all metrics have deteriorated except for FGD on feature space.

## 4 CONCLUSIONS AND DISCUSSION

In this paper, we propose a gesture generation system based on multimodal representation learning, where the considered modalities include text, audio and gesture. Each modality is projected into two different subspaces: modality-invariant and modality-specific. To learn the commonality among different modalities, an adversarial classifier based on gradient reversal layer is used. To capture the features of modality-specific representations, we adopt a modality reconstruction decoder. The gesture decoder utilizes all representations and audio rhythmic features to generate appropriate gestures. In subjective evaluation, our system is ranked fourth among the participating systems in human-likeness evaluation, and ranked seventh in appropriateness evaluation. Whereas, for appropriateness, the differences between our system and the five higher-ranked systems are not significant.

For appropriateness evaluation, whether there is a relationship between subjective evaluation and segmentation duration deserves to be investigated. The segments are around 8 to 10 seconds during evaluation[21]. We believe that a longer period of time (e.g. 20-30 seconds) might produce more pronounced and convincing appropriateness results.

There is room for improvement in this research. First, we only use data from one person to learn gesture due to unbalanced dataset issue. Such one-to-one mapping could produce boring and homogeneous gestures during inference. Second, the finger motions are not considered because of the low motion-capture quality. Such finger motions could be involved in the future if some data cleanup procedures could be conducted. Third, besides text and audio, more modalities (e.g. emotions, facial expressions and semantic meaning of gestures [16]) could be taken into consideration to generate more appropriate gestures.

## ACKNOWLEDGMENTS

This work is supported by Shenzhen Science and Technology Innovation Committee (WDZC20200818121348001), National Natural Science Foundation of China (62076144) and Shenzhen Key Laboratory of next generation interactive media innovative technology (ZDSYS20210623092001004).

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
