# OpenReview forum: "The ReprGesture entry to the GENEA Challenge 2022"
_ACM.org/ICMI/2022/Workshop/GENEA — GENEA Challenge & Workshop 2022 Mainproceeding_

### Official Review · Reviewer_CmXk · 2022-08-06
**Novel and interesting paper with a little unclarity.**

**Rating:** 7
**Confidence:** 4

**Review:**

The paper proposes a novel deterministic speech to gesture generation model by learning a modal-invariant and modal-specific features for different modalities used as inputs, i.e., audio, text, and seed poses. Ablation study showed that this method could improve the accuracy of the generated gestures.

Strengths:
1. Although the paper was not the best among proposed systems, the ablation study showed that the separation for modal-invariant feature and modal-specific feature were helpful to the accuracy of the generated gestures, i.e., a lower Hellinger distance and FGD on raw data space. This indicates that the proposed method is more effective to determine the gesture shape than purely mixing different modality features.
2. The using of WavLM, as a pre-trained neural network for audio feature extraction, improves the jerk and acceleration of the generated gestures, even though the pitch, energy and volume were used at the same time. This shows that prosodic features of audio, i.e., pitch and energy, are not enough for predicting the rhythmic movements in the gestures.

Potential issue:
The paper uses an autoencoder-like training scheme to help learn the hidden feature space. However, the reconstruction error was computed on the reconstructed results and the output of one of the hidden layers, whose parameters are being updated when training. This could lead to an undesired behavior that the encoder and decoder agree with each other while not considering the original input at all. A more common approach for training an autoencoder is to compute the reconstruction error on the original input or a transformation of the original input.

Weaknesses:
1. More details are necessary for understanding the domain learning. While the authors propose to extract modal-invariant features from modalities as in [5], they did not use the original similarity loss for these features. Instead, they propose to use domain learning to reach this goal. Although this could be one of the originalities of this paper, the authors did not thoroughly explain how domain learning works and how it can achieve a similar effect as central moment discrepancy (CMD) loss, or its potential advantages and disadvantages compared with CMD.
2. The authors claims that the reconstruction loss is used to ensure the hidden representations to capture the details. However, no ablation was provided for this. Thus, it is unclear that how much or if this loss is useful.
3. The authors did not use traditional feature extraction method on the audio such as mel-spectrogram, which is more common in the literature. The difference between using WavLM and basic features is unknown.

Questions:
1. The scales of coefficients for different loss terms ranges widely, e.g., alpha is 300 and epsilon is 0.1 in equation (5). How did the authors adjust the hyper-parameters and reach such a different scale?
2. Why did the author choose 3 seconds as the length for generation? Any reference for this?


**Nominate For A Reproducibility Award:**

No comment as the authors has not uploaded their code yet.

---

### Official Review · Reviewer_bVz8 · 2022-08-08
**The evaluation and sub-materials show good results.**

**Rating:** 9
**Confidence:** 4

**Review:**

The paper proposed a gesture generation method by incorporating representation learning into previously proposed network architectures. Six representations for audio, text, gesture (two per modality) are used. The method takes audio, text, and seed gestures as input and output is a sequence of gesture.

The paper is well organized and written but it is easier to read if put new lines after sub-sub-section titles.

The technical descriptions are well written and the experiments would be reproducible.

Some research, such as [10], have already addressed the importance of the audio or human-pose representations for gesture generation. So, the contribution is incremental, but having modality-mixed representations is somewhat novel.

The evaluation and sub-materials show good results. Having ablation study, also, help us to better understand the effects of proposed method. However, the effects of proposed representations were not very significant.

---

### Official Review · Reviewer_77RM · 2022-08-08
**The proposed method is somehow novel, but it lacks of ablation studies to validate the effects of using single modalities and cross-modality subspaces.**

**Rating:** 6
**Confidence:** 4

**Review:**

This paper presents a speech-driven gesture generation method based on multimodal representation learning. Each modality is projected to two distinct subspaces: modality-invariant and modality-specific. Gradient reversal layer based adversarial classifier and modality reconstruction decoders are used during training.
Average human-likeness ratings below 50% have been achieved, indicating that the proposed approach was not enough to generate natural gesture motions.

Ablation studies should be included to show the effects of modality-invariant only, or modality-specific only.
The video sample in the anonymous webpage shows an example of the generated motion.
I'm not aware of how the other dialogue segments used in the evaluation are, but in the sample, the target speaker is basically in listening mode, so that the instants the co-speech gestures are generated are very short.

---

### Decision · Program_Chairs · 2022-08-11

**Decision:**

Accept (Main proceeding)

**Comment:**

All three reviewers were in favour of accepting this paper. The chairs agree, and the paper is accepted to the ACM ICMI main proceedings

For the final (camera-ready) submission to ICMI, the chairs recommend the following changes, based on the reviews:

0) Replace the anonymous link to code, models, and video with a future-proof non-anonymous repository.

1) For the potential issue reviewer CmXk identified, please clarify whether or not a degenerate learning outcome, wherein the autoencoder ignores the input, is possible. Is it true that the reconstruction loss does not compare to the original input? If the reviewer is correct, comment on how you avoided these hypothetical degeneracy issues in practice. If this instead represents a misunderstanding on behalf of the reviewer, try to clarify the paper so that other readers do not risk getting the same mistaken impression.

2) Add a video example to the system webpage where the target speaker is speaking more than they are listening.

3) Consider adding details regarding the domain learning.

4) Comment on the reasoning behind using specifically 3 seconds for generation.

5) Comment on how the coefficients for the different loss terms were chosen/tuned.

6) Put newlines after subsubsection titles for readability, if space allows. This can possibly be accomplished using the \subsubsection command.